# Do's and Don'ts:
# Learning Desirable Skills with Instruction Videos

**Hyunseung Kim**[1,2]    **Byungkun Lee**[1]    **Hojoon Lee**[1]
**Dongyoon Hwang**[1]    **Donghu Kim**[1]    **Jaegul Choo**[1]
[1]KAIST, [2]KRAFTON
{mynsng,byungkun.lee,joonleesky,godnpeter,quagmire,jchoo}@kaist.ac.kr

## Abstract

Unsupervised skill discovery is a learning paradigm that aims to acquire diverse behaviors without explicit rewards. However, it faces challenges in learning complex behaviors and often leads to learning unsafe or undesirable behaviors. For instance, in various continuous control tasks, current unsupervised skill discovery methods succeed in learning basic locomotions like standing but struggle with learning more complex movements such as walking and running. Moreover, they may acquire unsafe behaviors like tripping and rolling or navigate to undesirable locations such as pitfalls or hazardous areas. In response, we present **DoDont** (Do's and Dont's), an instruction-based skill discovery algorithm composed of two stages. First, in instruction learning stage, DoDont leverages action-free instruction videos to train an instruction network to distinguish desirable transitions from undesirable ones. Then, in the skill learning stage, the instruction network adjusts the reward function of the skill discovery algorithm to weight the desired behaviors. Specifically, we integrate the instruction network into a distance-maximizing skill discovery algorithm, where the instruction network serves as the distance function. Empirically, with less than 8 instruction videos, DoDont effectively learns desirable behaviors and avoids undesirable ones across complex continuous control tasks. Code and videos are available at https://mynsng.github.io/dodont/

## 1   Introduction

Recent advancements in unsupervised pre-training methodologies have led to the creation of large-scale foundational models across diverse domains, including computer vision [3, 41] and natural language processing [7, 11, 22]. These methodologies exploit self-supervised learning objectives to extract meaningful representations without using explicit, supervised learning signals. In attempts to expand this paradigm to reinforcement learning, researchers have explored crafting self-supervised objectives to develop foundational policies capable of learning diverse behaviors without predefined reward signals, termed unsupervised skill discovery (USD) [1, 14, 9, 21, 43, 31, 24, 23, 27, 35, 38, 39].

Despite notable advancements in USD methodologies, acquiring foundational policies in environments with large state and action spaces (e.g., multi-jointed quadrupeds) remains a significant challenge. Two major issues arise when training agents with USD in these complex environments. First, the vast state and action spaces allow the agent to acquire a wide variety of behaviors. While learning simple behaviors like standing may be feasible, acquiring complex behaviors that require intricate joint movements, such as walking or running, can take an exceedingly long time. Second, agents can develop undesirable and risky behaviors during training, such as tripping, rolling, or navigating hazardous areas like pitfalls. These challenges raise a key question: *Is the purely unsupervised assumption of USD ideal for learning foundational policies in the real-world?*

38th Conference on Neural Information Processing Systems (NeurIPS 2024).

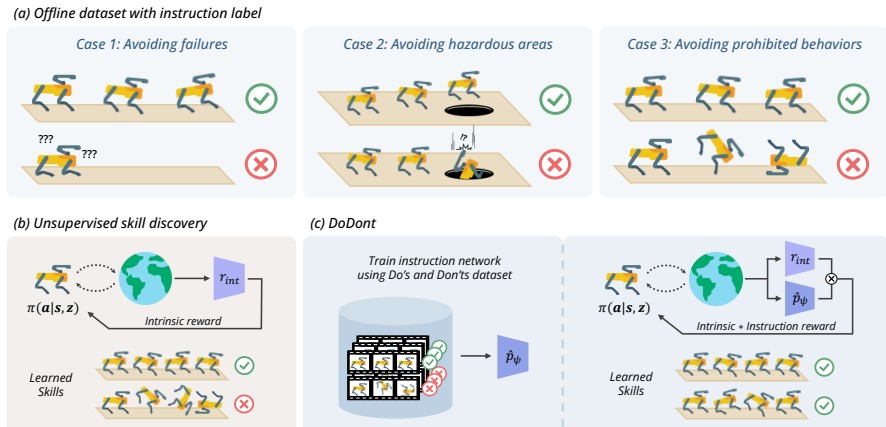

Figure 1: **(a)** The offline instruction video dataset includes videos of desirable behaviors (Do's) and undesirable behaviors (Don'ts). **(b)** Unsupervised skill discovery algorithms tend to learn undesirable behaviors. **(c)** In DoDont, an instruction network is first trained with the Do's and Don'ts videos to distinguish desirable and undesirable behaviors. Then, this instruction network adjusts the intrinsic reward of the skill discovery algorithm, promoting desirable skills while avoiding undesirable ones.

According to social learning theory, humans learn behaviors through both internal and external motivations [4, 5, 16]. Internally, they seek to perform diverse behaviors they haven't tried before. Externally, they adopt socially desirable behaviors and avoid those that are not. In the skill discovery algorithm, this can be mimicked by combining internal motivation (USD objective) with external motivation based on a human's intention. A straightforward way to incorporate external motivation in USD is to use a hand-designed reward function [26, 10]. However, hand-designing reward functions are not scalable for learning diverse and desirable behaviors for several reasons. One of the reasons is the complexity of designing an intricate reward signal for guiding a desired behavior, and another is the difficulty of balancing multiple behavioral intentions into a single reward function.

To address these challenges, we propose **DoDont**, a skill discovery algorithm that integrates USD objectives with intended behavioral goals. Instead of relying on a hand-designed reward, DoDont *learns a reward function from a small set of instruction videos* that demonstrate desirable and undesirable behaviors. Videos are chosen because they are inexpensive to collect and do not require action or reward labels [13, 46].

As illustrated in Figure 1, DoDont starts by collecting instruction videos of desirable (Do's) and undesirable behaviors (Don'ts). We then train an instruction network which assigns higher values to desirable behaviors and lower values to undesirable ones. This network re-weights the internal USD objective for the skill discovery phase. We utilize a distance-maximizing skill discovery algorithm as our main USD objective [35, 38, 39] where the instruction network serves as the distance metric.

To validate DoDont, we conduct experiments on various continuous control tasks that require complex locomotion (e.g., Cheetah and Quadruped [45]) or precise manipulation (e.g., Kitchen [17]). Our results show that with fewer than eight instruction videos, DoDont effectively learns complex locomotion skills (e.g., running quadruped), which are challenging to acquire with standard USD algorithms [39]. Moreover, our instruction network effectively captures human intentions better than hand-designed reward functions in balancing multiple behavioral objectives. Additionally, DoDont learns diverse skills while avoiding undesirable and unsafe behaviors (e.g., backflipping), which are difficult to prevent even with previous skill discovery algorithms which utilize prior knowledge [26].

## 2 Related work

### 2.1 Learning diverse behaviors without pre-defined task

Numerous unsupervised skill discovery (USD) methods have been developed to create foundational policies capable of learning a diverse behaviors without pre-defined task. The most com-

mon approach involves maximizing mutual information (MI) between states and skills ($I(S; Z) = D_{KL}(p(s,z)||p(s)p(z))$) [14, 43, 31, 9, 27]. This is typically done by using an auxiliary neural network $q_\theta(z|s)$ to estimate a lower bound of $I(S; Z)$ ($I(S; Z) \geq \mathbb{E}z, s[\log q_\theta(z|s)]$). This network acts as a skill discriminator, predicting the skill $z$ from the state $s$, which encourages the policy to create distinct trajectories for different skills and promotes learning a wide range of skills. However, these methods often struggle to learn complex, dynamic skills, as the MI objective can be met with simple, static skills, leaving agents unmotivated to explore more intricate behaviors [44, 23, 30, 31, 24].

To overcome this problem, distance-maximizing skill discovery (DSD) methods have been introduced [35, 38, 39]. Instead of maximizing MI between states and skills, DSD explicitly maximizes a predefined distance function of states $d : \mathcal{S} \times \mathcal{S} \to \mathbb{R}_0^+$. For instance, LSD [35] uses Euclidean distance between states to encourage agents to move further distances ($d_{\mathrm{LSD}} = ||s' - s||$). CSD [38] employs a density model based distance, promoting agents to visit less frequented states ($d_{\mathrm{CSD}} = -\log q_\theta(s'|s)$, where $q_\theta$ is the density model). METRA [39] defines the distance temporally, pushing agents to learn skills that are temporally far apart ($d_{\mathrm{METRA}}$ = minimum number of environment steps to reach $s'$ from $s$). DoDont is built upon the DSD framework, which will be elaborated in Sections 3 and 4.

## 2.2 Learning diverse behaviors with pre-defined task

In contrast to USD, there is a line of research focused on learning policies that exhibit diverse behaviors by integrating intrinsic motivation (e.g., MI-based rewards) with human intention, as the form of task-specific reward functions [10, 26, 55, 54] or using demonstration datasets [40]. While traditional RL methods typically aim to find a single optimal policy, these approaches aim to learn multiple policies $\pi_\theta(a|s, z_{i=1:N})$. This allows for multiple ways to solve given tasks, resulting in increased robustness to environmental changes and the development of policies with distinct characteristics.

SMERL [26] and DGPO [10] combine intrinsic and task rewards, maximizing their sum when task rewards exceed a given threshold. RSPO [55] iteratively finds novel policies by switching between task rewards and intrinsic diversity rewards, ensuring each new policy is distinct from previous ones. DOMiNO [54] addresses a constrained optimization problem by maximizing intrinsic diversity rewards while ensuring that all policies achieve sufficiently high performance (task rewards as constraints). ASE [40] utilizes a action-free demonstration dataset instead of a task reward function, minimizing a behavioral cloning loss while simultaneously maximizing a diversity intrinsic reward. DoDont also employs action-free demonstration datasets to extract human prior knowledge through an instruction network, pioneering the incorporation of both positive and negative behavior examples to guide learning towards desired behaviors and away from undesirable ones.

## 3 Preliminaries and problem setting

**Markov decision process and skill-conditioned policy.** Unsupervised skill discovery (USD) focuses on identifying a range of skills without pre-defined reward function. In this approach, we use a reward-free Markov Decision Process defined as $\mathcal{M} = (\mathcal{S}, \mathcal{A}, \mu, p)$. Here, $\mathcal{S}$ represents the state space, $\mathcal{A}$ is the action space, $\mu$ denotes the initial state distribution, and $p$ is transition dynamics function. We utilize a latent vector $z \in \mathcal{Z}$ and their associated policy $\pi(a|s, z)$, which we refer to as skills. To generate a skill trajectory (i.e., behavior), we first sample a skill $z$ from the prior distribution, denoted as $p(z)$, and then execute a trajectory using the skill policy $\pi(a|s, z)$. For the skill prior, we utilize a standard normal distribution to represent continuous skills by following previous works [35, 38, 39].

**Distance-maximizing skill discovery.** As discussed in Section 2.1, numerous distance-maximizing skill discovery methods have been proposed [35, 38, 39]. Specifically, DSD introduces the Wasserstein dependency measure (WDM) as a learning objective for unsupervised skill discovery (USD) [34].

$$I_{\mathcal{W}}(S; Z) = \mathcal{W}(p(s, z), p(s)p(z)) \tag{1}$$

Here, $\mathcal{W}$ is the 1-Wasserstein distance on the metric space $(S \times Z, d)$, where $d$ is a *distance metric*. Unlike MI-based skill discovery methods (maximizing $I(S; Z) = D_{KL}(p(s, z)||p(s)p(z))$) which does not explicitly motivate the agent to learn more complex behaviors (as discussed in Section 2.1), DSD maximizes the *distance-aware* $I_{\mathcal{W}}(S; Z)$. By doing so, DSD not only discovers a set of

distinguishable skills but also maximizes the distance $d$ between different states. By defining the distance function to encourage desired behaviors, DSD effectively induces the agent to learn these behaviors.

In practice, by leveraging the Kantorovich-Rubenstein duality [48, 34] under some simplifying assumptions, the following concise objective can be derived:

$$I_{\mathcal{W}}(S_T; Z) \approx \sup_{\|\phi\|_L \leq 1} \mathbb{E}_{p(\tau, z)} \left[ \sum_{t=0}^{T-1} \left( \phi(s_{t+1}) - \phi(s_t) \right)^\top z \right], \tag{2}$$

where $\phi : S \to \mathbb{R}^D$ is a function that maps states into $D$-dimensional space, $\|\phi\|_L$ denotes the Lipschitz constant for the function $\phi$ under the given distance metric $d$. Now, we can rewrite Equation 2 with an arbitrary distance function $d : S \times S \to \mathbb{R}_0^+$ as:

$$\sup_{\pi, \phi} \mathbb{E}_{p(\tau, z)} \left[ \sum_{t=0}^{T-1} \left( \phi(s_{t+1}) - \phi(s_t) \right)^\top z \right] \quad \text{s.t.} \quad \|\phi(x) - \phi(y)\|_2 \leq d(x, y), \quad \forall (x, y) \in S. \tag{3}$$

For a detailed derivation, we advise readers to refer to Section 4 of METRA [39]. It's important to note that a generic distance function, $d$, does not necessarily need to meet the criteria of a valid distance metric, such as symmetry or the triangle inequality. As shown in previous work [38], the original constraint in Equation 3 can be implicitly converted into one with a valid pseudometric, allowing us to use any non-negative function as a distance metric.

## 4 Do's and don'ts (DoDont)

Our goal is to integrate human intention into USD, where diverse policies are learned without predefined reward signals. The core concept of this work involves training an instruction network using video data to distinguish desired and undesired behvaiors (Section 4.1). Then, we set the trained instruction network as the distance metric in the distance-maximizing skill discovery framework (Section 4.2). Consequently, this will result in an agent that not only learns diverse behaviors but also incorporate human intention since the instruction network is set as the distance metric.

### 4.1 Instruction network

We aim to train an instruction network that assigns high values for desirable behaviors (Do's) and low values for undesirable behaviors (Don'ts). The distance metric in the WDM objective (in Equation 3) takes two states as input and should predict a non-negative value.

To meet this requirement, we first prepare videos depicting desirable behaviors (i.e., Do's) and videos illustrating behaviors to avoid (i.e., Don'ts) according to our intentions. We then label our video dataset by assigning $y = 1$ to adjacent state pairs $(s_t, s_{t+1})$ in Do's videos and $y = 0$ to those in Don'ts videos. This results in our Do's and Don't video dataset $\mathcal{D}_V$ which is comprised of triplets $(s_t, s_{t+1}, y)$. After preparing our video dataset $\mathcal{D}_V$, we use it to train our instruction network $\hat{p}_\psi$, which is designed to classify whether a given pair of adjacent states is from a Do's video or a Don'ts video. The training objective for the instruction network is a simple binary classification loss:

$$\mathcal{L}^{\text{Instruction}} = -\mathbb{E}_{(s_t, s_{t+1}, y) \sim D_V} \left[ y \log \hat{p}_\psi(s_t, s_{t+1}) + (1 - y) \log(1 - \hat{p}_\psi(s_t, s_{t+1})) \right]. \tag{4}$$

### 4.2 DoDont

Here, we integrate the learned instruction network into the online distance-maximizing skill discovery algorithm. As $\hat{p}_\psi$ is a non-negative function, we can directly use this instruction network as the distance metric of the WDM objective in the Equation 3 as:

$$\sup_{\pi, \phi} \mathbb{E}_{p(\tau, z)} \left[ \sum_{t=0}^{T-1} \left( \phi(s_{t+1}) - \phi(s_t) \right)^\top z \right] \quad \text{s.t.} \quad \|\phi(s) - \phi(s')\|_2 \leq \hat{p}_\psi(s, s'), \quad \forall (s, s') \in S_{adj}, \tag{5}$$

where $S_{adj}$ represents the set of adjacent state pairs. By assigning higher values to behaviors deemed desirable by humans (i.e., large distances) and lower values to behaviors that should be avoided, it guides the agent to learn skills that correspond to human intentions.

Then, following previous work [38, 39, 49], we can optimize Equation 5 with dual gradient descent, incorporating a Lagrange multiplier $\lambda$ and a small slack variable $\epsilon > 0$:

$$
\begin{aligned}
r^{\text{DoDont}} &:= (\phi(s') - \phi(s))^{\top} z, \\
\mathcal{J}^{\text{DoDont},\phi} &:= \mathbb{E}[(\phi(s') - \phi(s))^{\top} z] + \lambda \cdot \min(\epsilon, \hat{p}_{\psi}(s, s') - \|\phi(s) - \phi(s')\|), \\
\mathcal{J}^{\text{DoDont},\lambda} &:= -\lambda \cdot \mathbb{E}[\min(\epsilon, \hat{p}_{\psi}(s, s') - \|\phi(s) - \phi(s')\|)],
\end{aligned}
\tag{6}
$$

where $r^{\text{DoDont}}$ is the intrinsic reward for the latent conditioned policy, and $\mathcal{J}^{\text{DoDont},\phi}$ and $\mathcal{J}^{\text{DoDont},\lambda}$ are the objectives for $\phi$ and $\lambda$. We note that since the instruction network $\hat{p}_{\psi}$ is already trained on $D_V$, we keep $\hat{p}_{\psi}$ frozen for training stability.

However, one drawback of Equation 5 is that $r^{\text{DoDont}}$ can reflect instruction signals from $\hat{p}_{\psi}$ only after $\phi(s)$ has been sufficiently trained through $\mathcal{J}^{\text{DoDont},\phi}$. This indicates that in the early stages of training where $\phi(s)$ has not yet been sufficiently optimized, the agent may not properly receive instruction signals. Such delay in receiving instruction signals could potentially slow down the training process. Therefore, to ensure the agent receives instruction signals regardless of the training state of $\phi(s)$, we note that the following equation can be easily derived from Equation 5 :

$$
\sup_{\pi,\phi} \mathbb{E}_{p(\tau,z)} \left[ \sum_{t=0}^{T-1} \hat{p}_{\psi}(s, s') \left( \phi(s_{t+1}) - \phi(s_t) \right)^{\top} z \right] \quad \text{s.t.} \ \ \|\phi(s) - \phi(s')\|_2 \le 1, \ \ \forall (s, s') \in S_{adj},
\tag{7}
$$

For a detailed derivation, please refer to the Appendix B. Through this rephrasing, we now separate $\hat{p}_{\psi}$ from the training of $\phi(s)$. This allows the agent to receive a direct instruction signal, as $\hat{p}_{\psi}$ explicitly multiplies with the original intrinsic reward, independent of the training of $\phi(s)$. In our experiment, we found that this direct signal significantly enhances the agent's ability to learn diverse and desirable behaviors (Section 5.3). In addition, another advantage of Equation 7 is that it can be interpreted as simply multiplying instruction network $\hat{p}_{\psi}$ to the original learning objective function of METRA. This allows us to implement our algorithm by simply adding a single line of code on top of the METRA framework. We provide the pseudocode of DoDont Algorithm 1. Furthermore, our instruction network can be applied to zero-shot offline RL to learn diverse behaviors while prioritizing desirable behaviors within the offline unlabeled dataset. Detailed explanations and experiments are presented in Appendix A.

## 5 Experiments

### 5.1 Experimental setup

Our experiments aim to evaluate the effectiveness of the DoDont method in learning desirable behaviors while avoiding undesirable ones across various environments and instruction scenarios. We test DoDont in two contexts: complex locomotion tasks (Cheetah and Quadruped environments from the DeepMind Control (DMC) Suite [45]) and precise manipulation tasks (Kitchen environment [17, 33]) (Figure 2). For pixel-based inputs in the DMC environments, we colored the floors to help the agent be aware of its location from the pixel data, consistent with previous studies [36, 20, 39].

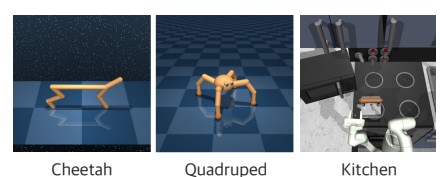

Cheetah     Quadruped     Kitchen

Figure 2: **Benchmark environments.**

For quantitative comparisons in locomotion tasks, we use two main metrics: (i) state coverage and (ii) zero-shot task reward. State coverage is measured by counting the $x$-axis coverage for Cheetah and $x, y$-axis coverage for Quadruped achieved by the learned skills at each evaluation epoch, following prior literature [39]. Zero-shot task reward evaluates the learned skills without task-specific training rewards to assess whether the agent efficiently learns or avoids specific behaviors. Our quantitative analysis uses four random seeds and provides 95% confidence intervals, represented by error bars or shaded areas. For each instruction scenario, we use eight videos (four "do" videos and four "don't" videos), unless otherwise specified. Additional details are elaborated in Appendix D.

To comprehensively evaluate DoDont's performance, we compare it against three types of baselines. First, to determine if an instruction network facilitates acquiring intended behavior, we compare DoDont with METRA [39], a state-of-the-art online unsupervised skill discovery algorithm. Second, to assess the effectiveness of our video-based intention network against a hand-designed reward function, we introduce METRA[†], a modified version of METRA that uses hand-designed reward functions as the distance metric. Finally, to verify if using an instruction network as a distance function effectively integrates our intention with skill discovery algorithms, we compare it to SMERL [26] and DGPO [10], which integrate task rewards in specialized ways. In this comparison, we use our intention network as the task reward function for both SMERL and DGPO. Details are in Appendix D.

## 5.2 Main experiment

In this section, our experiments are designed to answer the following research questions:

- How efficiently does DoDont learn complex behaviors in locomotion tasks? (Section 5.2.1)
- Does DoDont learn diverse behaviors while avoiding hazardous areas? (Section 5.2.2)
- Can DoDont learn diverse behaviors without learning unsafe behaviors? (Section 5.2.3)
- Can DoDont be applied to a manipulation environment? (Section 5.2.4)

### 5.2.1 How efficiently does DoDont learn complex behaviors in locomotion tasks?

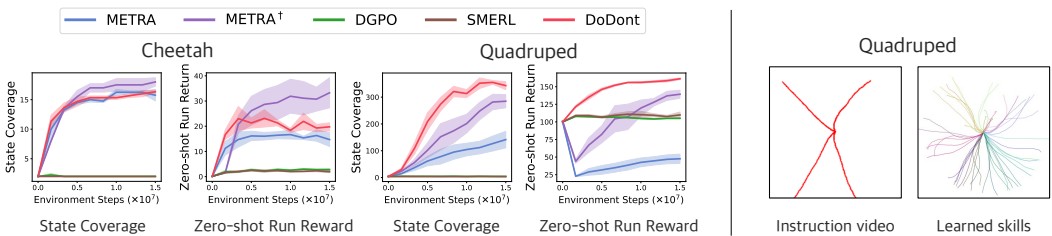

Figure 3: **Left**: State coverage and zero-shot task reward for Cheetah and Quadruped. **Right**: Visualization of Do videos in our instruction video dataset and learned skills by DoDont. We are able to observe that DoDont does not simply mimic instruction videos but extracts desirable behaviors (e.g., run) from the videos and learn diverse skills.

In this experiment, we aim to evaluate whether DoDont can effectively learn diverse complex behaviors by simply providing videos of desirable but complex behaviors. To accomplish this, we set videos of agents successfully running as Do's and random action videos as Don'ts for the Cheetah and Quadruped environment. As shown in Figure 3, DoDont achieves higher running rewards than METRA in both environments. Specifically, in the Quadruped environment, METRA primarily learns rolling movements for locomotion, while DoDont learns to run upright. Please refer to the project page videos for further details (link). Additionally, DoDont exhibits higher state coverage than METRA, since DoDont effectively learn running behaviors which allows the agent to cover longer distances whereas METRA only learn mediocre rolling behaviors.

We also observe that both SMERL and DGPO fail to learn diverse behaviors and only learns simple behaviors such as standing upright at the starting point with minimal movement. This limitation is likely due to their use of the MI objective, which appears insufficient for promoting diverse behavior learning in complex continuous control environments. METRA[†], a variant of METRA where we set the run task reward as the distance metric, is effective for simple environments with low-dimensional action spaces (Cheetah), but struggle in learning effective behaviors for environments with high-dimensional action spaces (Quadruped). We speculate that as the environment complexity increases, designing a single reward function which captures a variety of desirable behaviors becomes increasingly challenging.

An important point which we would like to emphasize is that DoDont only requires a total of eight videos to learn running behaviors. Despite the direction of each running video is limited (instruction videos in Figure 3), DoDont learns skill which run in all directions. This indicates that DoDont is not

merely imitating the behaviors in the Do's videos but discover a variety of behaviors which resemble those seen in the Do's videos. In addition, to further evaluate the meaningfulness of the learned behaviors, we conduct downstream task evaluation for each method in the Appendix A.

### 5.2.2 Does DoDont learn diverse behaviors while avoiding hazardous areas?

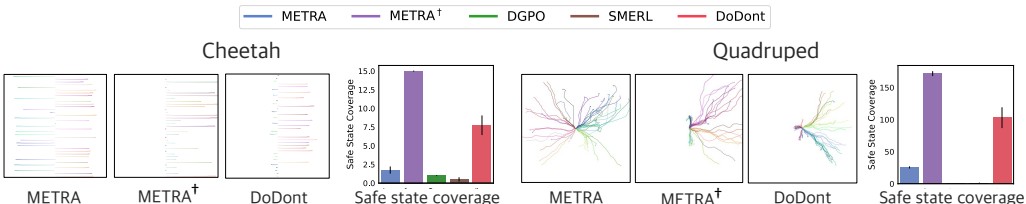

Figure 4: **Visualization and comparison of learned skills.** In both environments, the left side is hazardous and the right side is safe. Safe state coverage assesses the agent's ability to cover safe areas and avoid hazards.

In practical scenarios, it is crucial for agents to avoid hazardous areas. For instance, in navigation tasks, robots must avoid dangerous areas such as pitfalls or private property. However, traditional skill discovery methods, which aim to cover the entire state space, face difficulties in controlling the regions an agent explores. We now aim to evaluate whether the Don't videos in DoDont can reliably prevent agents from learning unwanted area navigation. To replicate real-world conditions, we perform experiments where we designate the left side of the environment as hazardous areas and the right side as safe areas. Thus, we use videos that move to the right as Do's and videos that move to the left as Don'ts, aiming to direct the agent away from hazardous zones and towards safer regions.

To evaluate the agent's ability to cover safe areas while avoiding hazardous ones, we introduced the concept of *safe state coverage*. This metric assigns a score of +1 for states in the safe area and -1 for states in the hazardous area. As illustrated in Figure 4, METRA without prior knowledge attempts to cover the entire state space, including hazardous areas, whereas DoDont effectively incorporates human intentions, avoiding dangerous areas while adequately covering the safe regions. We also observe that SMERL and DGPO fail to learn diverse skills, resulting in static behavior within the safe region, possibly due to the difficulty of learning diverse behaviors in pixel-based environments using the mutual information objective. Furthermore, METRA$^\dagger$ demonstrates superior performance in this experiment, likely because designing a reward function that incorporates human intentions is straightforward (assigning +1 for the safe region and 0 for the hazardous region).

### 5.2.3 Can DoDont learn diverse behaviors without learning unsafe behaviors?

In real-world scenarios, agents should also avoid learning risky behaviors. For instance, actions such as rolling on the ground or flipping upside down can potentially damage the robot. Therefore, in this experiment, we aim to investigate whether DoDont can learn a variety of desirable behaviors while avoiding certain unwanted behaviors. We trained our agent using running videos as desirable behaviors (Do's) and rolling or flipping videos as undesirable behaviors (Don'ts) for both Quadruped and Cheetah environments.

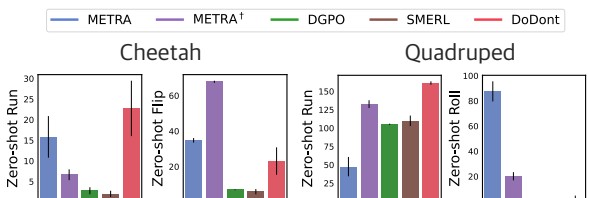

Figure 5: **Learning safe and diverse behaviors.** Zero-shot rewards assess how effectively each method learns desired behaviors while avoiding hazardous ones.

As shown in Figure 5, compared to METRA without any prior knowledge, DoDont effectively learns running behaviors while avoiding hazardous actions such as rolling or flipping using our instruction videos. Again, SMERL and DGPO fail to learn diverse behaviors, merely learning the simple behavior of standing upright at the starting point. For METRA$^\dagger$, we designed the reward function as $r_{\text{run}} - r_{\text{flip or roll}}$ to encourage running while avoiding flipping or rolling. We observe that METRA$^\dagger$ cannot fully encapsulate human intention in Cheetah environment. We believe this issue

arises from the entanglement of the task signals in the reward function which makes the agent difficult from discerning which type of behaviors are beneficial and which are not. To receive the run reward, the agent must move forward, but as the body moves, it sometimes triggers the flip reward, which penalizes the agent, hindering the agent from properly learning the run behavior. This illustrates that balancing multiple behavioral intentions is challenging, making the creation of hand-crafted reward functions difficult and unscalable.

### 5.2.4 Manipulation task

In this experiment, we aim to demonstrate that our intention network can be applied to a manipulation environment to learn multiple tasks. We use six target tasks following previous works [39, 33]. To effectively learn these six tasks, we use the D4RL dataset as Do's and random action videos as Don'ts. As shown in Figure 6, through our instruction network, DoDont successfully performs more tasks compared to METRA. SMERL and DGPO only learn two to three tasks, which we believe is due to the failure of the MI-based objective to capture diverse behaviors in pixel-based environments. For METRA[†], we use a hand-designed reward function by summing the rewards for all six tasks. Nevertheless, METRA[†] does not effectively learn the tasks, likely due to the entanglement of multiple behavioral signals within a single reward function.

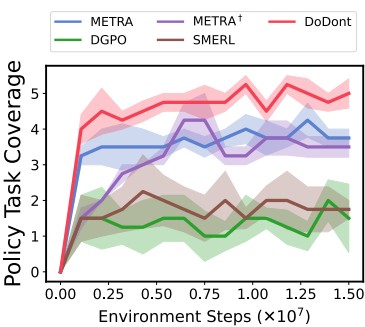

Figure 6: **Policy task coverage.** The number of tasks successfully completed by sampled skills.

### 5.3 Ablation Studies

### 5.3.1 Model components

We conduct an ablation analysis to demonstrate the importance of each component of our algorithm in achieving high performance. Specifically, we experimented with three variants: (i) using only one Do's video, (ii) optimizing Equation 5, referred to as delayed reward, and (iii) focusing solely on encouraging Do's behaviors. We tested these variants in a Quadruped environment, using run videos as Do's and random action videos as Don'ts.

**Impact of data quantity.** While we are already training with a relateively small amount of videos as our default setting (four Do's, four Don'ts), we also conduct an ablation study where we use only two videos (one Do, one Don't) to see how well our method performs under extremely limited data conditions. According to Figure 7, even with only one Do's video, the agent can mimic the running behavior, as evidenced by the zero-shot run reward. Remarkably, despite being exposed to just one video

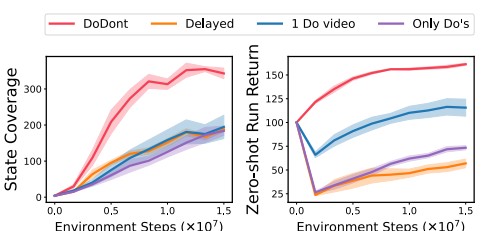

Figure 7: **Ablation study on model components.**

showing movement in a single direction, the agent is still able to learn movements in multiple directions; please refer to the project page videos (link).

**Direct reward signal.** To evaluate the effectiveness of Equation 7, we trained an agent using Equation 5. As shown in Figure 7, the direct instruction signal effectively captures the desired behavior from the early stages of training. In contrast, the delayed instruction signal initially fails to capture the desired behavior, as evidenced by drop in the zero-shot reward at the beginning. Consequently, the agent takes a long time to overcome before it can stand up again.

**Only encouraging Do's behaviors.** To further analyze the importance of penalizing Don'ts behaviors, we trained an agent only to encourage Do's behaviors. As illustrated in Figure 7, solely encouraging Do's behaviors fails to capture desirable behaviors. We speculate that without penalizing unwanted behaviors, the agent cannot avoid these undesirable actions, leading to a drop in the zero-

shot Run return at the beginning. Consequently, it takes a significant amount of time for the agent to learn the desired behaviors. Further details of this experiment in Appendix C.

### 5.3.2    The importance of utilizing instruction network as the distance metric

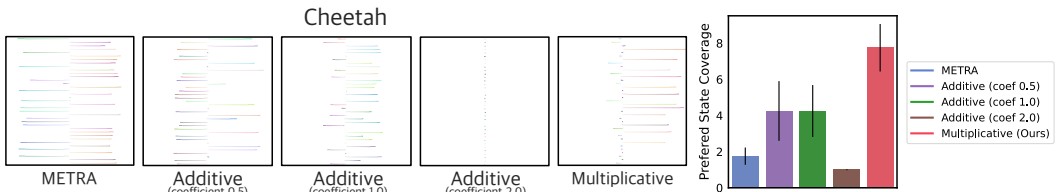

Figure 8: **Left:** Visualization of acquired skills, with the hazardous zone on the left and the safe zone on the right. **Right:** Quantitative comparison of each method.

In this study, we use the instruction network as the distance metric for the distance-maximizing skill discovery algorithm [39]. This setup is convenient because it only requires multiplying the instruction network with the intrinsic USD reward to train the policy. However, an alternative approach is to add the output of the instruction network with the intrinsic USD reward, similar to SMERL [26] and DGPO [10]. To compare these two approaches (multiplication and addition), we conducted experiments in the Cheetah environment following the setup from Section 5.2.2.

Our qualitative results, shown in Figure 8, indicate that none of the additive variants ($r_{\text{total}} = r_{\text{METRA}} + \alpha \hat{p}_\psi(s, s')$) match the performance of the multiplicative approach ($r_{\text{total}} = r_{\text{METRA}} \cdot \alpha \hat{p}_\psi(s, s')$). We attribute this to the fact that multiplying $\hat{p}_\psi(s, s')$ with the intrinsic reward function controls the scale of the total reward effectively since it directly impacts the influence of $r_{\text{METRA}}$ while adding $\hat{p}_\psi(s, s')$ indirectly impacts the influence of $r_{\text{METRA}}$. Although additive variants might achieve similar performance, they would likely require careful tuning of coefficients to balance the multiple reward components.

## 6    Conclusions, limitations, and future work

In this paper, we introduce DoDont, an instruction-based skill discovery algorithm designed to combine human intention with unsupervised skill discovery. We empirically demonstrate that our instruction network enables the agent to effectively learn desirable behaviors and avoid undesirable ones across a range of realistic instruction scenarios.

However, our study faces certain limitations. Specifically, in this research, we trained the instruction network using in-domain video data, which is not a readily available resource in real-world applications. We propose that training the instruction model with general, in-the-wild video data represents a scalable and compelling direction for future investigation.

Despite these constraints, DoDont exhibits promising results even with a limited video dataset. Moreover, we believe that videos are a cost-effective form of data for representing behaviors as they do not require action and reward labels. Thus, preparing data is both practical and feasible. On the other hand, as video generation models advance, several recent works have utilized these models as simulators [50, 51, 12, 6]. There is potential to generate Do's and Don'ts videos with these models, which could allow for the use of generated data without the need for additional data collection.

As final remarks, in unsupervised skill discovery, there are many excellent prior works that have succeeded in learning diverse behaviors without any supervision, data, or prior knowledge. However, these methods often struggle to acquire complex behaviors and may learn hazardous behaviors, making them unsuitable for direct real-world application. Therefore, we believe that the next step in unsupervised reinforcement learning focuses on leveraging minimal supervision or data (i.e., resources that are easily obtainable in the real world) to become scalable and applicable in complex control tasks and practical for the real world.

## Acknowledgments

We are grateful to Seohong Park for providing insightful discussions and valuable feedback. This work was supported by Institute for Information & communications Technology Promotion(IITP) grant funded by the Korea government(MSIT) (No.RS-2019-II190075 Artificial Intelligence Graduate School Program(KAIST)), and the National Research Foundation of Korea (NRF) grant funded by the Korea government (MSIT) (No. NRF-2022R1A2B5B02001913).

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

# Appendix

# A Additional Experiments

## A.1 Improving zero-shot offline RL with instruction videos

Recently, HILP [37] has introduced a novel framework for pre-training generalist policies that learn diverse behaviors from task-agnostic unlabeled datasets [52]. HILP can be considered as an offline variant of METRA, as both methodologies focus on learning temporal distance-preserving representations to facilitate the acquisition of diverse behaviors. Similar to our previous application in METRA, we can easily integrate our instruction network into HILP's intrinsic reward function. The pseudocode for the offline version of DoDont is provided in Algorithm 2. Intuitively, by utilizing our instruction network, the policy is trained to learn diverse behaviors while prioritizing desirable behaviors and avoiding hazardous ones within the offline dataset.

In this experiment, we further investigate the effectiveness of our method when applied to an offline zero-shot reinforcement learning (RL) approach that learns diverse behaviors from task-agnostic, unlabeled datasets. Zero-shot RL is divided into two phases: reward-free pretraining with an offline dataset and maximizing an arbitrary reward function provided at test time without additional training. We primarily compared our results with HILP [37], FB [47], and FDM [37, 47], the leading methods in offline zero-shot RL. We conducted experiments in four environments (Walker, Cheetah, Quadruped, and Jaco) using two ExORL datasets [52] collected by APS [31] and RND [8]. We followed the experimental protocol outlined in the HILP paper [37], where detailed information is available in Appendix D.

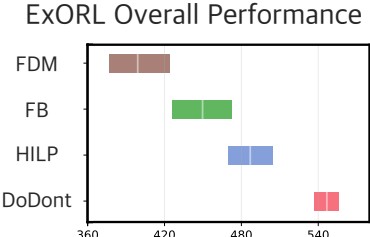

Figure 9: **Aggregated performance.** The overall results are aggregated over 4 environments, 4 tasks, 2 datasets, and 4 seeds (i.e., 128 values in total).

During the pre-training phase, we employed four downstream task videos as Do's and four random action videos as Don'ts for each environment. This strategy aims to prioritize the learning of task-relevant behaviors. We use the interquartile mean (IQM) [2] for overall aggregation to assess performance. As illustrated in Figure 9, DoDont is superior over other methods by a large margin in effectively capturing downstream task behaviors from a large, task-agnostic, unlabeled dataset. We would like to note that although DoDont benefits from downstream task relevant data, it significantly outperforms baselines while utilizing only a small set of instruction videos. Detailed results are in Appendix E.

## A.2 Downstream task performance

In this section, we employed a hierarchical controller that selects (frozen) learned skills to maximize the downstream task reward to evaluate the meaningfulness of the learned behaviors. As depicted in Figure 10, behaviors learned through DoDont showed higher performance than ME-TRA, indicating that DoDont effectively learned useful behaviors that can be applied to downstream tasks.

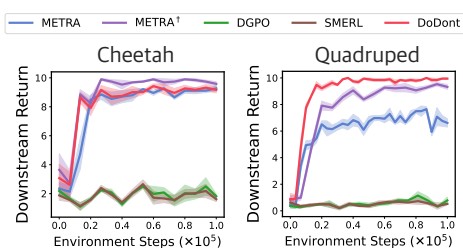

Figure 10: **Downstream task performance.**

## B  Derivation of equation 7

We first start with Eq. 5:

$$\sup_{\pi,\phi} \mathbb{E}_{p(\tau,z)} \left[ \sum_{t=0}^{T-1} (\phi(s_{t+1}) - \phi(s_t))^\top z \right] \quad \text{s.t.} \quad \|\phi(s) - \phi(s')\|_2 \leq \hat{p}_\psi(s, s'), \quad \forall(s, s') \in S_{adj}, \quad (8)$$

Let scaled state function $\tilde{\phi}(s) := \frac{\phi(s)}{\hat{p}_\psi(s,s')}$. Then, we can transform the constraint term in Eq. 8 as follows ($\hat{p}_\psi(s, s') \geq 0$ since it is the output value of binary classification network),

$$\sup_{\pi,\phi} \mathbb{E}_{p(\tau,z)} \left[ \sum_{t=0}^{T-1} (\phi(s_{t+1}) - \phi(s_t))^\top z \right] \quad \text{s.t.} \quad \|\tilde{\phi}(s) - \tilde{\phi}(s')\|_2 \leq 1, \quad \forall(s, s') \in S_{adj}, \quad (9)$$

By replacing $\phi(s)$ with $\tilde{\phi}(s) \cdot \hat{p}_\psi(s, s')$ in Eq. 9, we derive

$$\sup_{\pi,\phi} \mathbb{E}_{p(\tau,z)} \left[ \sum_{t=0}^{T-1} \hat{p}_\psi(s_t, s_{t+1}) \left( \tilde{\phi}(s_{t+1}) - \tilde{\phi}(s_t) \right)^\top z \right] \quad \text{s.t.} \quad \|\tilde{\phi}(s) - \tilde{\phi}(s')\|_2 \leq 1, \quad \forall(s, s') \in S_{adj}.$$
$$(10)$$

## C  Implementation Details of DoDont

Here, we provide further details regarding the implementation of DoDont. Our experiments are based on two different codebases: METRA[1] [39] for online learning and HILP[2] [37] for offline learning. Our experiments run on NVIDIA RTX 3090 GPUs, with each run taking no more than 28 hours.

### C.1  Data curation

We now explain how the action-free Do's and Don't video dataset we used as our human intention dataset were collected. For DMC tasks, we first trained expert Soft Actor-Critic (SAC) [18] policies based on proprioceptive states with the ground truth task reward for over 2 million environment steps and generate Do videos by deploying the expert policy in the environment and saving the visual observations rendered by the simulator. In addition, we set the ExORL dataset [52] as our Don't videos for DMC tasks since the ExORL dataset is mainly composed of transitions generated via undesirable random behavior. On the other hand for the Kitchen environment, we set the the D4RL dataset [15] as our Do's video dataset and additionally gather videos of random action rollouts and set them as our Don't videos. The number of videos used for each environment can be found in Table 1.

Table 1: **The number of videos used for each environment.**

| Environment | # of Do's videos | # of Don'ts videos |
|---|---|---|
| Online DMC | 4 | 4 |
| Kitchen | # of videos in D4RL dataset | 10 |
| Offline Zero-shot DMC | 12 videos per task | # of videos in ExORL dataset |

### C.2  Instruction network

For the instruction network, we use four convolutional layers for the backbone and three fully connected layers for the prediction head. To improve stability, we bound the output using the Tanh function, following previous work [29]. Additionally, we employ simple random shift augmentation, a widely used data augmentation technique. The Adam optimizer [25] is employed with a learning rate of $1 \times 10^{-4}$ and a batch size of 1024.

---

[1]https://github.com/seohongpark/METRA
[2]https://github.com/seohongpark/HILP

## C.3 Skill learning

We implemented DoDont based on two algorithms: METRA [39] and HILP [37]. We used the hyperparameters from these algorithms without modification. Additionally, DoDont introduces only one extra hyperparameter, which is the coefficient for the instruction network. The complete list of hyperparameters can be found in Table 2, 3.

## C.4 Pseudocode of DoDont

We present the pseudocode for the online version of DoDont in Algorithm 1 and the offline version of DoDont in Algorithm 2.

---
**Algorithm 1:** Do's and Don'ts (Online)

---
1  **Initialize** instruction network $\hat{p}_\psi(s, s')$, video dataset $\mathcal{D}_V$
2  **for** *number of epochs* **do**
3  $\quad$ Sample batch $(\sigma^0, \sigma^1, y)$ from video dataset $\mathcal{D}_V$
4  $\quad$ Update $\hat{p}_\psi(s, s')$ to minimize $\mathcal{L}^{\text{Instruction}}$ in ( 4)
5  **Initialize** policy $\pi(a|s, z)$, representation function $\phi(s)$, Lagrange multiplier $\lambda$, replay buffer $\mathcal{D}$
6  **for** *number of policy epochs* **do**
7  $\quad$ **for** *number of episodes per epochs* **do**
8  $\quad\quad$ Sample skill $z \sim p(z)$
9  $\quad\quad$ Sample trajectory $\tau$ with $\pi(a|s, z)$ and add to replay buffer $\mathcal{D}$
10 $\quad$ Update $\phi(s)$ to maximize $\mathbb{E}_{(s,z,s')\sim\mathcal{D}}[(\phi(s') - \phi(s))^\top z + \lambda \cdot \min(\epsilon, 1 - \|\phi(s) - \phi(s')\|)]$
11 $\quad$ Update $\lambda$ to minimize $\mathbb{E}_{(s,z,s')\sim\mathcal{D}}[\lambda \cdot \min(\epsilon, 1 - \|\phi(s) - \phi(s')\|)]$
12 $\quad$ Update $\pi(a|s, z)$ using SAC with reward $r(s, z, s') = \hat{p}_\psi(s, s')(\phi(s') - \phi(s))^\top z$

---

---
**Algorithm 2:** Do's and Don'ts (Offline)

---
1  **Initialize** representation $\phi(s)$
2  **Initialize** policy $\pi(a|s, z)$
3  **while** *not converged* **do**
4  $\quad$ Sample $(s, s', g) \sim \mathcal{D}$
5  $\quad$ Train $\phi(s)$
6  **Train** instruction network $\hat{p}_\psi(s, s')$
7  **while** *not converged* **do**
8  $\quad$ Sample $(s, a, s') \sim \mathcal{D}$ and $z \sim \mathbb{S}^{D-1}$
9  $\quad$ Compute intrinsic reward $r(s, z, s')$
10 $\quad$ Train $\pi(a|s, z)$ with weighted reward $\hat{p}_\psi(s, s')r(s, z, s')$

---

# D Experimental details

## D.1 Environments

**Benchmark environments.** In this work, we perform experiments with the Cheetah and Quadruped environment from DMC [45] and a pixel-based version of the Kitchen environment [17, 33]. Following previous studies [39, 36, 20], for the pixel-based DMC environments we use gradient-colored floors which allows the agent to infer its location from pixels. For the Kitchen environment, we utilize the same camera settings as LEXA [33].

For Sections 5.2.1, 5.2.3, and 5.3, we use state information as input for both the policy and the critic. For Sections 5.2.2 and 5.2.4, we use pixel data as input for the policy and the critic. The reason for this experimental setup is that in pixel-based environments with gradient-colored floors, the discriminator might exploit the background color as a shortcut to distinguish between different states rather than observing the agent's embodiment. This makes it challenging to incorporate behavioral intentions. However, without colored floors, METRA cannot learn diverse behaviors. Therefore,

Table 2: **Hyperparameters used in Online DoDont.** We adopt default hyperparameters from METRA [39], introducing only one additional hyperparameter.

| Hyperparameter | Value |
|---|---|
| Learning rate | 0.0001 |
| Optimizer | Adam [25] |
| # episodes per epoch | 8 |
| Encoder | CNN [28] |
| # hidden layers | 2 |
| # hidden units per layer | 1024 |
| Minibatch size | 256 |
| Target network smoothing coefficient | 0.995 |
| # gradient steps per epoch | 50 (state-based DMC), 200 (pixel-based DMC), 100 (Kitchen) |
| Replay buffer size | $10^6$ (state-based DMC), $3 \times 10^5$ (pixel-based DMC), $10^5$ (Kitchen) |
| Entropy coefficient | auto-adjust [19] (DMC), 0.01 (Kitchen) |
| $\epsilon$ | 0.001 |
| initial $\lambda$ | 30 |
| Latent dimension | 4 (continuous) (DMC), 24 (discrete) (Kitchen) |
| Instruction network coefficient | 2 |

Table 3: **Hyperparameters used in Offline DoDont.** We adopt default hyperparameters from HILP [37], introducing only one additional hyperparameter.

| Hyperparameter | Value |
|---|---|
| # gradient steps | $10^6$ |
| Learning rate | 0.0005 ($\phi$), 0.0001(others) |
| Optimizer | Adam [25] |
| Minibatch size | 1024 |
| MLP dimensions | (512, 512) ($\phi$), (1024, 1024, 1024) (others) |
| TD3 target smoothing coefficient | 0.99 |
| Latent dimension | 50 |
| # state samples for latent vector inference | 10000 |
| Successor feature loss | Q loss on {Quadruped, Jaco}, vector loss (others) |
| Hilbert discount factor | 0.96 (Walker), 0.98 (others) |
| Hilbert expectile | 0.5 |
| Hilbert target smoothing coefficient | 0.995 |
| Instruction network coefficient | 3 (Walker), 2 (others) |

for Sections 5.2.1, 5.2.3, and 5.3, where behavioral intentions are included, we use state-based environments and use non-colored default pixel as a human intentions. Notably, all experiments use videos (i.e., pixel data) as a human intentions.

For offline zero-shot RL, we use four environments (Walker, Cheetah, Quadruped, and Jaco) and two different ExORL datasets [52] (APS [31], RND [8]). Although experiments in the HILP [37] paper is based on four ExORL datasets (APS [31], RND [8], Proto [53], APT [32]), due to limited computational resources, we performed experiments only with the APS and RND ExORL dataset since these two dataset are where HILP achieved the best performance.

**Metrics.** For the state coverage in DMC, we count the number of $1 \times 1$ $x - y$ bins occupied by any of the target trajectories for Quadruped and 1-sized $x$ bins for the Cheetah. In the Kitchen environment, we count the number of predefined tasks achieved by any of the target trajectories. We use the same six predefined tasks as LEXA [33] and METRA [39]: Kettle (K), Microwave (M),

Light Switch (LS), Hinge Cabinet (HC), Slide Cabinet (SC), and Bottom Burner (BB). Policy state coverage is computed using 48 deterministic trajectories with 48 randomly sampled skills at the current epoch.

For offline zero-shot RL performance, we follow the method outlined in HILP [37]. We sample a small number of $(s, a, s')$ tuples from the dataset, compute the optimal latent $z^*$ with respect to the test-time reward function, execute the corresponding policy $\pi(a|s, z^*)$, and compute the return value. We use all the hyperparameters from HILP without modification.

**Downstream tasks.**  We utilize two downstream tasks: QuadrupedGoal and CheetahGoal, primarily following METRA [39]. For each task, the objective is to reach a target spot randomly sampled from $[-7.5, 7.5]^2$ for QuadrupedGoal and $[-10.0, 10.0]^2$ for CheetahGoal. The agent receives a reward of 10 upon reaching within a radius of 3 units from the target spot. We train a hierarchical high-level controller on top of the frozen skill policy. The high-level controller selects a skill $z$ for every $K = 50$ environment steps, and the pre-trained skill policy $\pi(a|s, z)$ executes the same skill $z$ for those $K$ steps. We use PPO [42] for discrete skills and SAC [18] for continuous skills, maintaining all hyperparameters as in METRA [39].

### D.2  Implementation details of baseline algorithms

In this work, we use four baselines: METRA [39], METRA$^\dagger$, DGPO [10], and SMERL [26]. For METRA, we used the open-source METRA codebase and retained the original hyperparameters. METRA$^\dagger$ is a variant of METRA that incorporates hand-designed reward functions as a distance metric. As described in the main paper, we manually designed the reward function and rescaled it to be greater than zero since it is used as a distance metric. Like the DoDont method, we multiplied METRA's intrinsic reward by this hand-designed reward function.

DGPO introduces an intrinsic reward function $r^{\text{int}} = \min_{z' \neq z} \log \frac{q(z|s')}{q(z|s') + q(z'|s')}$. With this intrinsic reward, we use our instruction network as a human intention. Total reward function for DGPO is $r = \hat{p}_\psi(s, s') + \alpha \min_{z' \neq z} \log \frac{q(z|s')}{q(z|s') + q(z'|s')}$, where $\alpha$ is the balancing coefficient.

SMERL uses an intrinsic reward defined as $r^{\text{int}} = \mathbb{1}_{R_\mathcal{M}(\pi_\theta) \geq R_\mathcal{M}(\pi_\mathcal{M}^*) - \epsilon} \tilde{r}$. Like DGPO, we employed our instruction network as a human intention. To train SMERL, we first trained a baseline SAC agent with our instruction network and considered the maximum return achieved by the trained SAC agent as the optimal return $R_\mathcal{M}(\pi_\mathcal{M}^*)$. We used the same $\tilde{r} = q_\phi(z|s)$ as described in the original paper. The total reward function for SMERL is $r = \hat{p}_\psi(s, s') + \alpha \mathbb{1}_{R_\mathcal{M}(\pi_\theta) \geq R_\mathcal{M}(\pi_\mathcal{M}^*) - \epsilon} q_\phi(z|s)$, where $\alpha$ is the balancing coefficient.

In Section 5.3, we introduce "Only Do's," a variant of our algorithm that exclusively promotes desired behaviors. We achieve this by eliminating the penalties for undesirable behaviors. Specifically, we define $p_{\text{dos}}(s, s')$ as follows:

$$p_{\text{dos}}(s, s') = \begin{cases} \hat{p}_\psi(s, s') & \text{if } \hat{p}_\psi(s, s') \geq 0.5 \\ 0.5 & \text{if } \hat{p}_\psi(s, s') < 0.5 \end{cases} \tag{11}$$

This approach ensures that the probability $p_{\text{dos}}(s, s')$ never falls below 0.5, thereby encouraging only the desired behaviors.

### D.3  License

- Environment
    - Deepmind Control suite: We used the publicly available environment code in [39, 45][3].
    - Kitchen: We used the publicly available environment code in [39, 15][4].
- Dataset
    - ExORL: We used the publicly available dataset code in [52][5].

---

[3]https://github.com/seohongpark/METRA
[4]https://github.com/seohongpark/METRA
[5]https://github.com/denisyarats/exorl

# E Full results

## E.1 Zero-shot RL

Table 4: **Full results on the zero-shot RL performance.** The table shows the zero-shot RL performance averaged over four seeds in each setting. We adopted the results from HILP [37]

| Dataset | Environment | Task | FDM | FB | HILP | DoDont |
|---|---|---|---|---|---|---|
| APS | Walker | Flip | $426 \pm 68$ | $334 \pm 178$ | $573 \pm 37$ | $506 \pm 70$ |
| | | Run | $248 \pm 23$ | $388 \pm 27$ | $348 \pm 14$ | $355 \pm 46$ |
| | | Stand | $865 \pm 77$ | $824 \pm 54$ | $883 \pm 42$ | $953 \pm 22$ |
| | | Walk | $634 \pm 91$ | $842 \pm 105$ | $862 \pm 31$ | $930 \pm 34$ |
| | Cheetah | Run Backward | $116 \pm 116$ | $250 \pm 135$ | $373 \pm 72$ | $383 \pm 19$ |
| | | Run | $360 \pm 17$ | $251 \pm 39$ | $316 \pm 21$ | $290 \pm 44$ |
| | | Walk | $396 \pm 287$ | $683 \pm 267$ | $939 \pm 55$ | $984 \pm 2$ |
| | | Walk Backward | $982 \pm 2$ | $980 \pm 3$ | $985 \pm 2$ | $985 \pm 2$ |
| | Quadruped | Jump | $707 \pm 30$ | $757 \pm 52$ | $623 \pm 149$ | $721 \pm 30$ |
| | | Run | $481 \pm 4$ | $474 \pm 33$ | $411 \pm 62$ | $489 \pm 6$ |
| | | Stand | $961 \pm 20$ | $949 \pm 30$ | $797 \pm 117$ | $965 \pm 4$ |
| | | Walk | $578 \pm 145$ | $583 \pm 12$ | $605 \pm 75$ | $568 \pm 55$ |
| | Jaco | Reach Bottom Left | $12 \pm 18$ | $14 \pm 12$ | $88 \pm 41$ | $61 \pm 17$ |
| | | Reach Bottom Right | $28 \pm 20$ | $24 \pm 7$ | $48 \pm 24$ | $60 \pm 24$ |
| | | Reach Top Left | $21 \pm 16$ | $23 \pm 17$ | $49 \pm 18$ | $88 \pm 31$ |
| | | Reach Top Right | $34 \pm 40$ | $17 \pm 15$ | $51 \pm 32$ | $64 \pm 22$ |
| RND | Walker | Flip | $415 \pm 20$ | $548 \pm 94$ | $563 \pm 136$ | $644 \pm 33$ |
| | | Run | $295 \pm 70$ | $409 \pm 15$ | $401 \pm 30$ | $436 \pm 22$ |
| | | Stand | $821 \pm 84$ | $866 \pm 120$ | $800 \pm 61$ | $894 \pm 31$ |
| | | Walk | $476 \pm 259$ | $811 \pm 52$ | $855 \pm 34$ | $872 \pm 32$ |
| | Cheetah | Run Backward | $107 \pm 26$ | $183 \pm 83$ | $262 \pm 53$ | $257 \pm 6$ |
| | | Run | $177 \pm 62$ | $153 \pm 41$ | $187 \pm 55$ | $226 \pm 10$ |
| | | Walk | $494 \pm 122$ | $636 \pm 291$ | $823 \pm 141$ | $944 \pm 15$ |
| | | Walk Backward | $652 \pm 274$ | $677 \pm 85$ | $843 \pm 184$ | $955 \pm 1$ |
| | Quadruped | Jump | $758 \pm 98$ | $642 \pm 36$ | $556 \pm 101$ | $684 \pm 16$ |
| | | Run | $491 \pm 7$ | $436 \pm 26$ | $393 \pm 42$ | $494 \pm 35$ |
| | | Stand | $971 \pm 11$ | $797 \pm 72$ | $810 \pm 97$ | $859 \pm 79$ |
| | | Walk | $601 \pm 82$ | $642 \pm 202$ | $542 \pm 32$ | $753 \pm 58$ |
| | Jaco | Reach Bottom Left | $53 \pm 20$ | $18 \pm 11$ | $19 \pm 20$ | $33 \pm 14$ |
| | | Reach Bottom Right | $38 \pm 14$ | $29 \pm 12$ | $18 \pm 17$ | $29 \pm 31$ |
| | | Reach Top Left | $36 \pm 19$ | $45 \pm 16$ | $8 \pm 10$ | $33 \pm 15$ |
| | | Reach Top Right | $44 \pm 24$ | $22 \pm 7$ | $5 \pm 4$ | $9 \pm 9$ |

# F Broader impact

Our research has a broader impact on the safety and reliability of unsupervised skill discovery (USD) algorithms. By incorporating instruction-based learning to differentiate between desirable and undesirable behaviors, our method significantly lowers the risk of agents developing unsafe behaviors, such as tripping or navigating hazardous areas. This improvement enhances the practical usability of USD in real-world scenarios where safety is critical, especially in applications like autonomous vehicles and robotic manipulation. However, a major issue with USD remains its sample inefficiency due to the need for extensive simulations.

