# OpenReview forum: "Do's and Don'ts: Learning Desirable Skills with Instruction Videos"
_NeurIPS.cc/2024/Conference — NeurIPS 2024 poster_

### Official Review · Reviewer_DB1p · 2024-07-06

**Soundness:** 4
**Presentation:** 4
**Contribution:** 2
**Rating:** 7
**Confidence:** 3

**Summary:**

The paper introduces "DoDont", an instruction-based skill discovery algorithm designed to learn desirable behaviors and avoid undesirable ones through unsupervised skill discovery (USD). The method uses instruction videos to train an instruction network that distinguishes between desirable (Do’s) and undesirable (Don’ts) behaviors. This network adjusts the reward function of the skill discovery algorithm to encourage desired behaviors. The authors validate their approach through experiments in complex continuous control tasks, demonstrating that DoDont can effectively learn desirable behaviors with minimal instruction videos.

**Strengths:**

- The integration of instructional videos into the USD framework is innovative and addresses the challenge of learning desirable behaviors without predefined reward signals.
- This paper stands out for its practical value, offering a USD algorithm that effectively learns meaningful and complex behaviors rather than merely generating skills that are variations of simple actions/jittering.
- The paper provides thorough experimental validation on three tasks, showing that DoDont outperforms state-of-the-art methods in learning complex and desirable behaviors.
- The presentation, writing and clarity of the paper are great.

**Weaknesses:**

- The instruction network is trained using in-domain video data, which might not always be readily available in real-world scenarios, but can be fairly easy to obtain.

**Questions:**

- What are the specific characteristics of the instruction videos required for effective training? For example, do they need to be of a certain length, resolution, or context?
- How does the quality and clarity of the instructional videos impact the performance of the DoDont algorithm?
- In Section 2.2, alongside DOMiNO, I think you should cite [1] and [2] that also balance a trade-off between intrinsic reward and task reward using constrained optimization.
- I am not clear on the state space for each tasks. Do you use exclusively visual pixel inputs or do you combine the visual input stream with proprioceptive data? For example, the labeled consecutive states $(s_t, s_{t+1})$ are only pixel values or combined with proprioceptive data?

[1] Skill-Conditioned Policy Optimization with Successor Features Representations\
[2] Quality-Diversity Actor-Critic: Learning High-Performing and Diverse Behaviors via Value and Successor Features Critics

**Limitations:**

Yes, the authors adequately addressed the limitations.

---

> ### Author Rebuttal · Authors · 2024-08-06
>
> Dear reviewer DB1p,
>
> Thank you for your insightful feedbacks and the positive support. We have provided a detailed response to your concerns below. If you have any further comments, please let us know.
>
> >**Question 3.1**
> The instruction network is trained using in-domain video data, which might not always be readily available in real-world scenarios, but can be fairly easy to obtain.
>
> We agree. Using in-the-wild video datasets is an important direction for future work, which would expand our method's real-world applicability. This limitation has been discussed in detail in Section 6 of our paper; please refer to that section for further details.
>
> >**Question 3.2**
> What are the specific characteristics of the instruction videos required for effective training? For example, do they need to be of a certain length, resolution, or context\
> How does the quality and clarity of the instructional videos impact the performance of the DoDont algorithm?
>
> The most critical characteristic of instruction videos for effective training is visual consistency with the training environment. Specifically, the resolution and quality of the videos should closely match those of the training environment. If there's a significant discrepancy between the pixel characteristics in the instructional videos and the training environment, the network's predictive accuracy diminishes, resulting in less accurate human-aligned behaviors.
>
> Regarding length, it's worth noting that our method trains the instruction network by sampling two consecutive frames from the videos. Therefore, the total duration of each video is not a crucial factor in the training process.
>
> >**Question 3.3**
> In Section 2.2, alongside DOMiNO, I think you should cite [1] and [2] that also balance a trade-off between intrinsic reward and task reward using constrained optimization.
>
> Thank you very much. We will ensure that they are included in the final version.
>
> >**Question 3.4**
> I am not clear on the state space for each tasks. Do you use exclusively visual pixel inputs or do you combine the visual input stream with proprioceptive data? For example, the labeled consecutive states (st, st+1)  are only pixel values or combined with proprioceptive data?
>
> We apologize for the ambiguity regarding the state space for each task. We employ only pixel inputs for the instruction network in both state-based and pixel-based environments. For the policy and critic networks, we utilize state inputs in state-based environments and pixel inputs in pixel-based environments.
>
> We will include a comprehensive explanation in the final version of the paper.

---

> > ### Comment · Reviewer_DB1p · 2024-08-08
> >
> > Thank you for your detailed rebuttal. I will maintain my score.

---

### Official Review · Reviewer_oJKL · 2024-07-13

**Soundness:** 3
**Presentation:** 4
**Contribution:** 3
**Rating:** 7
**Confidence:** 4

**Summary:**

Unsupervised skill discovery is an RL task to learn interesting behaviors without rewards from environments. However, since there is no specification of desired behavior either, a lot of learning is wasted on acquiring skills that people may not be interested in eventually. The paper studies a setting where a few demonstration video are provided as a minimal specification of desirable skills. Then the proposed method can train a GAN like loss as a distance measure for unsupervised skill discovery, so the skill learned is more desirable. Experiments on dmlab are shown with interesting analysis.

**Strengths:**

1. The paper is well written
2. The setup is well-motivated. It's indeed interesting to see something between complete unsupervised RL & imitation learning.
3. A lot of the ablation gives the readers good insights about the proposed method, which I find enjoyable to learn.

**Weaknesses:**

1. The idea of combining unsupervised RL with some form of task specification is not new. In the very early days of unsupervised RL, people already add intrinsic reward and extrinsic reward together. I hope the authors could justify their novelty

2. While the authors discusses USD, I feel a lot of works in RL exploration are not mentioned. e.g. Curiosity [1]. There are other works about using very high level information as guidance, such as DeepMimic [2]. I am curious to see whether the proposed method can be combined with general USD methods?



[1] Curiosity-driven Exploration by Self-supervised Prediction, Deepak Pathak, Pulkit Agrawal, Alexei A. Efros, Trevor Darrell
[2] DeepMimic: Example-Guided Deep Reinforcement Learning of Physics-Based Character Skills

**Questions:**

1. What's the camera angle for "Quadruped"? Clearly the acquired skill is more diverse than instruction video as shown in figure 3. I am wondering how, because intuitively those trajectories are equally out of distribution.

**Limitations:**

The paper discusses the limitation of scaling to in-the-wild videos. I think this is reasonable address. No ethics concerns.

---

> ### Author Rebuttal · Authors · 2024-08-06
>
> Dear Reviewer oJKL,
>
> Thank you for your valuable feedback on our paper. We have carefully considered your concerns and would like to address them as follows. Please let us know if you have further questions or feedbacks.
>
> >**Question 2.1**
> The idea of combining unsupervised RL with some form of task specification is not new. In the very early days of unsupervised RL, people already add intrinsic reward and extrinsic reward together. I hope the authors could justify their novelty
>
> We appreciate the reviewer's comment about the existing work in combining intrinsic rewards with extrinsic rewards. While it's correct that there exist early approaches combined intrinsic and extrinsic rewards, DoDont's main contributions are in:
>    1. Modeling "human-aligned behaviors" (i.e., how to design human-aligned extrinsic rewards)
>    2. Integrating this reward into multiple-behavior learning (i.e., how to combine extrinsic and intrinsic rewards effectively)
>
> Previous approach such as ICM [1] or SMERL [2] added intrinsic rewards with hand-crafted extrinsic rewards. However, as shown in Figure 6, designing hand-crafted extrinsic rewards for multiple human-aligned behaviors is very difficult. DoDont overcomes this challenge by:
>    1. Training a "human-aligned instruction network" using easily obtainable instruction videos
>    2. Employing this network as a "distance-metric in distance-maximizing skill discovery", thus facilitating the learning of diverse human-aligned skills
>
> This approach effectively induces multiple human-aligned behaviors, allowing for diverse behavior learning without unsafe actions. We will emphasize these aspects in our revised manuscript to clarify our contribution.
>
> >**Question 2.2**
> While the authors discusses USD, I feel a lot of works in RL exploration are not mentioned. e.g. Curiosity [1]. There are other works about using very high level information as guidance, such as DeepMimic [2]. I am curious to see whether the proposed method can be combined with general USD methods?
>
> For unsupervised RL methods (e.g., ICM[1], RND[3]), it's possible to combine these with DoDont (e.g., reward = r_ICM + r_DoDont). This would likely result in improved exploration while maintaining **single** human-aligned policy. However, our paper focuses on learning **multiple** human-aligned behaviors with a limited instruction dataset, we chose the USD algorithm (DSD) as our baseline.
>
> For unsupervised skill discovery methods, as outlined in Section 4, DoDont involves two key components: (1) modeling human-aligned behaviors (extrinsic rewards) and (2) combining these extrinsic rewards with intrinsic rewards. The instruction network learned in step (1) can also be applied to other methods like DIAYN [4] and CIC [5] by combining our instruction-network reward with their intrinsic rewards (e.g., reward = r_DIAYN + r_DoDont). However, given that MI-based algorithms such as DIAYN and CIC face inherent pessimistic exploration challenges [6], we believe our approach would be most effective when applied to distance-maximizing skill discovery algorithms.
>
> >**Question 2.3**
> What's the camera angle for "Quadruped"?
>
> For DMC Quadruped, we employ the standard camera angle as depicted in Figure 2.
>
> >**Question 2.4**
> Clearly the acquired skill is more diverse than instruction video as shown in figure 3. I am wondering how, because intuitively those trajectories are equally out of distribution.
>
> This likely stems from the generalization capabilities of the instruction network. Instruction network is trained to classify behaviors, categorizing moving upright as '1' and rolling on the floor (i.e., random actions) as '0'. Consequently, even when faced with previously unseen directions of movement, the network tends to assign a high value to upright movements.
>
> [1] Curiosity-driven Exploration by Self-supervised Prediction., Pathak et al., ICML 2017\
> [2] One Solution is Not All You Need: Few-Shot Extrapolation via Structured MaxEnt RL., Kumar et al., NeurIPS 2020\
> [3] Exploration by Random Network Distillation., Burda et al., ArXiv 2018\
> [4] Diversity is All You Need: Learning Skills without a Reward Function., Eysenbach et al., ICLR 2019\
> [5] CIC: Contrastive Intrinsic Control for Unsupervised Skill Discovery., Laskin et al., ArXiv 2022\
> [6] Learning More Skills through Optimistic Exploration., Strouse et al., ICLR 2022\

---

> > ### Comment · Reviewer_oJKL · 2024-08-09
> >
> > Thank you for your rebuttal. I think an average score of slightly above 6 is fair to this paper. I will maintain my rating of 7.

---

### Official Review · Reviewer_AtB3 · 2024-07-16

**Soundness:** 3
**Presentation:** 3
**Contribution:** 2
**Rating:** 5
**Confidence:** 3

**Summary:**

This paper proposes a method, DoDont, to avoid hand-crafting reward functions in unsupervised skill discovery. DoDont first learns a reward function from labelled instruction videos that discriminates desired and undesired behaviors, and then use the reward function in unsupervised skill discovery. The authors evaluate the method in several experimental settings and find DoDont can learn more diverse and safer skills than the baselines.

**Strengths:**

Clear motivation that hand-crafting rewards in unsupervised skill discovery is tedious. The proposed method makes sense and is presented fairly clearly.

**Weaknesses:**

I think the major weakness here is using a implicit reward function instead of explicit hand-crafted rewards has been explored before while those baselines are missing in the paper. DoDont learn the reward on some labelled instruction videos. There are several other approaches of automatic reward design in the past, for example, using a LLM [1, 2] or a vision-language model [3], which the authors should compare DoDont to and explain the advantages of DoDont.

[1] Ma, Yecheng Jason, et al. "Eureka: Human-level reward design via coding large language models." arXiv preprint arXiv:2310.12931 (2023).

[2] Kwon, Minae, et al. "Reward design with language models." arXiv preprint arXiv:2303.00001 (2023).

[3] Fan, Linxi, et al. "Minedojo: Building open-ended embodied agents with internet-scale knowledge." Advances in Neural Information Processing Systems 35 (2022): 18343-18362.

**Questions:**

1. According to Appendix C.1, the Do videos for DMC tasks is collected from a policy that's trained on ground truth reward functions. Does this mean DoDont still needs hand-crafting "ground-truth" rewards for a new task? If so then DoDont is not solving the motivating problem of USD that it needs hand-crafted rewards.
2. From the same section, it seems the Don't videos are collected from random action rollouts. Will these be enough for learning a reward that can teach the model to avoid hazardous behaviors? For example, walking into a hole is a bad behavior, but random action rollouts probably won't touch this kind of trajectory because random actions won't make an agent walk in the first place. Therefore, the reward module hasn't seen such bad behaviors during training and probably can't learn to assign a low reward value to this behavior.

**Limitations:**

See above.

---

> ### Author Rebuttal · Authors · 2024-08-06
>
> Dear reviewer AtB3,
>
> Thank you for your constructive comments. We have provided a detailed response to your comments below. Please let us know if you have further questions or feedbacks.
>
> >**Question 1.1**
> I think the major weakness here is using a implicit reward function instead of explicit hand-crafted rewards [...], using a LLM [1, 2] or a vision-language model [3], which the authors should compare DoDont to and explain the advantages of DoDont.
>
> We are grateful for your constructive feedback and the chance to elucidate aspects of our research.
>
> We apologize for any ambiguity in defining the scope and objectives of our study.
> We would like to emphasize the distinct scope and objectives of our research in comparison to the studies [1,2,3].
> In our study, DoDont addresses the challenges inherent in unsupervised skill discovery (USD), specifically the inefficiencies and potential hazards of learning undesired behaviors. Our approach leverages instruction videos to guide the discovery of **diverse, desirable, and safe behaviors**, by applying *distance metric* learned from instruction videos to distance-maximizing skill discovery.
>
> It's important to note that our approach is distinct from the objectives of the studies cited in the review [1, 2, 3], which predominantly focus on crafting a reward function for a **single** task using LLMs or VLMs. As such, a direct empirical comparison might not be readily feasible.
>
> While our current approach doesn't directly utilize LLMs or VLMs, we recognize their potential as a viable "distance metric".
> This could result in more human-aligned behaviors, for instance, by capturing the significant semantic distance between states such as "jumping to the right" and "remaining seated".
>  Nevertheless, adopting these models would introduce notable computational challenges and resource demands, which we outline as follows:
>
> - **Eureka [1]** employs complex prompt tuning and multiple iterations of RL policy training, which can be resource-intensive.
> - **Reward Design with Language Models [2]** necessitates frequent queries to LLMs during environment step, leading to high inference costs.
> - **Minedojo [3]** depends on extensive image-text datasets for training VLMs, which are not always feasible to compile.
>
> Given these considerations, and the constraints of the rebuttal period, we conducted preliminary tests using LLMs to establish distance metrics, specifically using [2] for the 5.2.2 experimental setting to avoid hazardous areas.
>
> We would like to note that [2] employs the 'text-davinci-002' GPT-3 model as its LLM. However, querying the LLM at every environment step incurs high inference costs, so we opted for the open-source LLM, Llama-3-8B. To generate the distance metric, we used prompts such as:
>
> "You will describe the consecutive x, y, z positions of a given robot moving on a plane. The robot's consecutive positions are given as: [x1, y1, z1], [x2, y2, z2]. If the x-coordinate value increases, the robot moves to the right. In this case, output the scalar value 1.0. If the x-coordinate value decreases, the robot moves to the left. In this case, output the scalar value 0.0."
>
> As shown in the attached PDF file under "Global Response," the agent effectively incorporates human intention, avoiding hazardous areas while adequately covering the safe regions. However, even with an open-source LLM, inferring the 8B model at each environment step incurs high computational time and cost. Moreover, while creating prompts for basic tasks like "move to the right" is relatively simple, developing prompts for more complex desired behaviors becomes increasingly challenging. In comparison, DoDont operates efficiently with minimal data, typically requiring only one to four pairs of instructional videos. We believe that our methodology remains a simple and practical solution.
>
> We appreciate your feedback and will incorporate this information into the appendix.
>
> >**Question 1.2**
> According to Appendix C.1, the Do videos for DMC tasks is collected from a policy that's trained on ground truth reward functions. [...] If so then DoDont is not solving the motivating problem of USD that it needs hand-crafted rewards.
>
> We wish to clarify that instruction videos does not necessitate the availability of task-specific reward functions. These instruction videos can be acquired through several means.
> For instance, as outlined in Section 5.2.4 (kitchen environment), they can be obtained from human-collected datasets.
> Additionally, demonstration videos on platforms such as YouTube (e.g., Minecraft videos) can serve as instruction videos.
> For humanoid robots, teleoperation techniques [1,2] offer another viable source for these videos.
> These diverse examples illustrate that the DoDont algorithm is not constrained to the pre-defined task-specific rewards but can operate effectively when provided with suitable behavioral videos.
>
> [1] HumanPlus: Humanoid Shadowing and Imitation from Humans., Fu et al., arXiv 2024\
> [2] Open-TeleVision: Teleoperation with Immersive Active Visual Feedback., Cheng et al., arXiv 2024
>
> >**Question 1.3**
> From the same section, it seems the Don't videos are collected from random action rollouts. [...] probably can't learn to assign a low reward value to this behavior.
>
> Addressing the reviewer's concern, it is important to clarify that our methodology for learning diverse skills while avoiding hazardous zones and unsafe behaviors does not merely use random actions as Don’t videos. Specifically, in our experimental setup outlined in Section 5.2.2, where we identify the left areas as hazardous, our Don't videos depict leftward movements rather than random actions (refer to lines 243-244). In Section 5.2.3, which deals with avoiding unsafe behaviors, we use videos depicting rolling or flipping as Don’t videos (refer to lines 267-268).
>
> We acknowledge the omissions in Appendix C.1 and will provide a thorough explanation in the finalized version of the paper.

---

> > ### Comment · Reviewer_AtB3 · 2024-08-11
> > **Thank you for the response**
> >
> > I would like to thank the authors for the thoughtful response. I think my concerns about data are resolved. For the weakness, while the authors and Reviewer oJKL consider the related papers [1, 2, 3] having different objectives and thus not comparable, I still think at least one or two of them should serve as baselines to DoDont. The task DoDont is trying to solve is avoiding undesired and hazardous behaviors in unsupervised skill discovery and the main contribution of DoDont is avoiding hand-crafting reward functions in this task by introducing an implicit reward function that is learned on instruction videos. Since the main contribution of DoDont is removing hand-crafting reward function in this process and the listed papers also propose methods to replace hand-crafted rewards with automatic reward design, DoDont should compare to these methods to analyze what is the best practice for reward design in the area of unsupervised skill discovery. I see that the authors have some preliminary attempts on using LLM to generate rewards which I think is valuable and I encourage the authors to include more of such comparisons or ablations in a revised version of the paper. Given the additional comparison to LLM baselines and the concerns on data being addressed, I've increased my score to 5.

---

> ### Comment · Reviewer_oJKL · 2024-08-09
>
> I strongly agree with the authors that [1,2,3] in the review are of a distinct scope and shall not constitute the sole reason to reject this paper.
>
>
> [1] Ma, Yecheng Jason, et al. "Eureka: Human-level reward design via coding large language models." arXiv preprint arXiv:2310.12931 (2023).
>
> [2] Kwon, Minae, et al. "Reward design with language models." arXiv preprint arXiv:2303.00001 (2023).
>
> [3] Fan, Linxi, et al. "Minedojo: Building open-ended embodied agents with internet-scale knowledge." Advances in Neural Information Processing Systems 35 (2022): 18343-18362.

---

### Author Rebuttal · Authors · 2024-08-06

**Response to All Reviewers (General Response)**

We deeply appreciate the thoughtful feedback and valuable suggestions from all three reviewers. R1, R2, and R3 correspond to reviewer AtB3, reviewer oJKL, and reviewer DB1p, respectively.

The reviewers highlighted the following strengths in our submission:

- The main idea is intuitive and well-motivated (R1, R2, R3).
- The proposed method was evaluated against several methods in various domains, yielding promising empirical results (R2, R3).

However, the reviewers also suggested conducting several key experiments to enhance our paper:

- Comparison with previous automatic reward design approaches (R1).
- Justification of our novelty (How our approach differs from existing studies that combine extrinsic rewards with unsupervised RL) (R2).

We hope our responses address all the reviewers’ concerns, and we welcome any additional comments and clarifications.

Additionally, we have attached a PDF file containing the qualitative and quantitative evaluation results for R1.

---

### Public Comment · ~Dian_Cheng1 · 2025-09-26
**No opensources codes on website https://mynsng.github.io/dodont/**

Dear Author,

I read your paper and found it very interesting. However, when I visited the website (https://mynsng.github.io/dodont/), I couldn’t find any code related to the article. Might you have forgotten to upload the code? This seems inconsistent with the statement in the abstract on OpenReview and in the final version of the paper: “Code and videos are available at [https://mynsng.github.io/dodont/].”

Do you plan to update the website and release the code in the near future?

---

### Decision · Program_Chairs · 2024-09-25

**Decision:**

Accept (poster)

**Comment:**

This paper presents an unsupervised skill discovery algorithm that addresses safe and effective exploration through the concept of desired and undesired behaviors learned from a few example videos. The empirical evaluations are comprehensive, demonstrating complex continuous control skills.

The reviewers initially expressed concerns, but after the rebuttal phase, they agreed to accept the paper, contingent on additional experimental results and clarification regarding comparisons to prior work.

The authors are encouraged to address the following points discussed during rebuttal:
* provide comprehensive comparisons to some LLM- and VLM-based reward designing approaches
* discuss the challenges associated with acquiring undesirable behavior videos, especially in real-world settings
* address the limited application to real-world environments (e.g. pixel or high-dimensional observations)
* moderate the claims regarding safety guarantees during training